# From Sensor to Cloud: An IoT Network of Radon Outdoor Probes to Monitor Active Volcanoes

**DOI:** 10.3390/s20102755

**Published:** 2020-05-12

**Authors:** Luca Terray, Laurent Royer, David Sarramia, Cyrille Achard, Etienne Bourdeau, Patrick Chardon, Alexandre Claude, Jérôme Fuchet, Pierre-Jean Gauthier, David Grimbichler, Jérémy Mezhoud, Francis Ogereau, Richard Vandaële, Vincent Breton

**Affiliations:** 1Laboratoire de Physique de Clermont, Université Clermont Auvergne, CNRS/IN2P3, 63000 Clermont-Ferrand, France; luca.terray@clermont.in2p3.fr (L.T.); laurent.royer@clermont.in2p3.fr (L.R.); david.sarramia@clermont.in2p3.fr (D.S.); cyrille.achard@clermont.in2p3.fr (C.A.); patrick.chardon@clermont.in2p3.fr (P.C.); alexandre.claude@clermont.in2p3.fr (A.C.); richard.vandaele@clermont.in2p3.fr (R.V.); 2Laboratoire Magmas et Volcans, Université Clermont Auvergne, CNRS/INSU, 63000 Clermont-Ferrand, France; pierre-jean.gauthier@uca.fr; 3Université Clermont Auvergne, Mésocentre, DSI, 63000 Clermont-Ferrand, France; etienne.bourdeau@uca.fr (E.B.); dje.emploi@free.fr (J.F.); david.grimbichler@uca.fr (D.G.); jeremy.mezhoud@uca.fr (J.M.); francis.ogereau@uca.fr (F.O.)

**Keywords:** IoT, radon, wireless sensor networks, sensors, LoRaWAN, data lake, volcano monitoring

## Abstract

While radon in soil gases has been identified for decades as a potential precursor of volcanic eruptions, there has been a recent interest for monitoring radon in air on active volcanoes. We present here the first network of outdoor air radon sensors that was installed successfully on Mt. Etna volcano, Sicily, Italy in September 2019. Small radon sensors designed for workers and home dosimetry were tropicalized in order to be operated continuously in harsh volcanic conditions with an autonomy of several months. Two stations have been installed on the south flank of the volcano at ~3000 m of elevation. A private network has been deployed in order to transfer the measurements from the stations directly to a server located in France, using a low-power wide-area transmission technology from Internet of Things (IoT) called LoRaWAN. Data finally feed a data lake, allowing flexibility in data management and sharing. A first analysis of the radon datasets confirms previous observations, while adding temporal information never accessed before. The observed performances confirm IoT solutions are very adapted to active volcano monitoring in terms of range, autonomy, and data loss.

## 1. Introduction and State-of-the-Art

### 1.1. Radon Monitoring on Active Volcanoes

Volcanic eruptions are still today extremely difficult to predict, and geological monitoring mostly relies on the search for precursors. A precursor is an alerting phenomenon that takes place sufficiently prior to the occurrence of an eruption. The recent dramatic events at Stromboli, one of the world’s most active volcanoes, highlight that volcanic activity is not always announced by precursory signals that we are able to observe. It is thus important to improve our knowledge of precursory manifestations, both by identifying new types of monitoring parameters and by better understanding how well-known monitoring parameters (such as seismicity, ground deformation, or gas composition) relate to volcanic activity.

As early as 1975, soil radon emissions have been identified as a tracer of volcanic activity (see, for instance, [1]). Since then, many studies have been conducted on Mount Etna (see, for instance, [2,3] and references therein) and on many other volcanoes [4] to assess the potential of radon as precursor of volcanic activity. Radon gas emissions in soils have been monitored continuously, and correlations with volcanic or tectonic events have been repeatedly observed [5]. However, because of the complexity of the radon transport mechanism, these correlations need more investigations [6]. 

Beyond soil radon emissions, very few studies have investigated radon concentrations in the air in volcanic areas. One of these studies has documented the activity of radon in outdoor air in 25 locations of the Mt. Etna volcano region [7]. These preliminary results did not confirm elevated values close to tectonic faults. On the other hand, at three points close to the summit, activities up to 93 Bq/m^3^ were found. More recently, radon activity in diluted volcanic gases of Mt. Etna was measured by passive dosimetry [8]. This observation strongly suggests for the first time that the plume is enriched in radon. Moreover, covariations of radon activity in the air and volcanic activity have been noticed and indicate that radon activity in magmatic gas plumes could also be an interesting parameter to monitor, as was previously proposed by theoretical modeling [9]. Finally, this study has also revealed very high radon activities (up to 8 kBq/m^3^) in the air in a few zones of the Mt. Etna central crater that are characterized by enhanced soil fracturation and degassing. This suggests that in very active zones where soil radon probes cannot be permanently installed because of high temperature and acidity, soil radon emissions could still be monitored by measurements in the air. To further investigate the potential of radon activity in the air as a tool to monitor active volcanoes, continuous measurements at high frequencies are obviously needed.

In this paper, we present the preliminary deployment and test of a network of monitoring stations designed to provide continuous measurements of radon activity in the air in a volcanic environment.

### 1.2. Sensors for Radon Monitoring in the Air

While radon activity measurement in diluted volcanic gas is a relatively unexplored field of research, indoor airborne radon has been routinely monitored for decades, especially for public health issues. The health risks related to radon and its progeny through inhalation or ingestion are well-known and have been studied for many years (see, for instance, the handbook provided by the World Health Organization [10]). Outdoor airborne radon measurements are also routinely performed in the context of environmental surveillance of radioactive waste disposal facilities [11]. Therefore, many technologies have been designed to measure radon in the air (for a recent technological review, see [12]). Now, cost-effective Internet of Things (IoT) systems for monitoring indoor radon gas concentrations are also available [13].

Measuring radon concentration in the air on an active volcano is very challenging for several reasons. First of all, radon activity levels found in the air are much smaller than activity levels in soil gases because of important dilution. On Mt. Etna, airborne radon activity measured on the volcano flanks are typically of a few tens of Bq/m^3^ [7]. Contrastingly, radon activity in soil gases in active areas is generally higher than 1000 Bq/m^3^ and can reach several tens of kBq/m^3^ (see [3,5], for instance). Since the radon activity to be measured in the air is low to moderate, the response time of any radon measurement instrument will be much higher than in the context of soil gas studies, and as a consequence, time resolution will be poorer. Very low-activity (<1 Bq/m^3^) continuous radon measurements with excellent time resolutions (<1 h) have been developed for decades by atmospheric scientists [14]. However, such instruments require the aspiration of high volumes of air and, thus, need a significant power supply, which is not easily compatible with field conditions on active volcanoes. Indeed, active volcanoes are extreme environments for both meteorological and volcanological reasons. For instance, during the winter, Mt. Etna summit craters are hardly accessible due to snowfalls, and the operation of solar panels is very irregular and uncertain. Even in summer, the access to relevant measurement sites is only possible by foot. Moreover, when put in contact with volcanic gases and ash, instruments have to be protected against corrosion. Finally, in very active zones, eruptive hazards seriously threaten any installation; as a consequence, low-cost and easily replaceable equipment is generally preferred. For all these reasons, deployed sensors must be thought to be resistant to hard outdoor conditions, autonomous energetically for several months, and relatively compact and low cost.

These three expectations (good time resolution, high autonomy/robustness, and low size/cost) are not easily concealed, and the choice of a measuring device is always a compromise between very precise and sophisticated instruments with very low expected lifetimes and less precise/more basic, easy to replace sensors with longer lifetimes. Among potential commercial devices for airborne radon monitoring, Alpha Guard radon monitors (Saphymo GmbH, Frankfurt, Germany) have already been used in a volcanic context [7]. Such systems collect air and pass it into an ionization chamber where radon and its daughters can be measured. Of great interest for outdoor field campaigns [15], these devices are not autonomous and robust enough for being left on the volcano during the winter (between 10 and 30 days). 

Another physical option to measure radon is by the electrical collection of its daughters onto solid-state silicon detectors (see, for instance, [16]). Since this measurement technique needs a lower bias voltage (a few volts) compared to ionization chambers, it can be operated with very low power consumption and long autonomy using small batteries. On the other hand, they usually require longer integration periods, since measurement chambers are limited in volume by the achievable size of silicon photodiodes. Among others, the ÆR+ radon probe (Algade, Bessines-sur-Gartempe, France; [17]) uses this measurement principle with a typical sensitivity of 0.05 hits per hour per Bq/m^3^ (equivalent to 20 Bq/m^3^ per hits per hour). Its one-year autonomy makes it well-fitted for long measurement campaigns such as those considered in a volcanic context. Moreover, ÆR+ radon probes have been tested in a calibration chamber with varying humidity and proved to be very adapted to outdoor radon monitoring [18]. However, the probe is not tropicalized nor protected for operation in a volcanic environment. 

In this paper, we describe the tropicalization of ÆR+ radon probes in order to make them suitable for deployment at high altitudes (>3000 m) and in acidic environments on Mt. Etna Volcano.

### 1.3. IoT Network Technologies toward Large-Scale Deployment of Small Sensors on Active Volcanoes

Volcano monitoring implies a data transfer from measuring stations to an observatory where the information can be treated and analyzed in real time. In a case of pending eruption, data transfer has to be as fast and as robust as possible in order to allow the earliest alert as possible. In a perspective of monitoring oriented toward real-time crisis management, the data transfer step has a predominant importance, since it will condition the quantity of information available to experts. What is more, the complexity of volcanic systems often requires geographically distributed measurements rather than punctual records in order to draw relevant scientific conclusions. For this reason, sensors are deployed in large networks covering important geographical areas, making data transfer even more challenging. The recent development of small sensor technologies relevant to many domains of volcano monitoring (mostly geodesy [19] and gas composition [20]) is opening the way to the deployment of networks with unprecedented density. Such deployments require innovative data transfer solutions able to manage efficiently a very high number of stations.

In this global context, volcano monitoring has been clearly identified as an area of particular interest for applying the Internet of Things technology. As early as 2004, a US research project deployed a small test wireless sensor network on the Tungurahua Volcano in Central Ecuador [21]. A wireless sensor network was developed for temporary or permanent monitoring of seismic tremors and deployed successfully on a suspended bridge for measuring its vibrations [22]. More recently, a wireless sensor network was deployed on the Masaya Volcano, Nicaragua, using Libelium technology (Libelium, Zaragoza, Spain; [23]). Wireless networks are also considered to deploy thermometers in order to measure soil temperatures in volcanic areas [24].

In most cases, sensor data are collected at site levels on gateways using the low-power wide area LoRa network technology. This technology is intended to achieve a long-range transmission of limited amounts of information between so-called nodes with very low power consumption. As a consequence, its usage has spread rapidly to a large portfolio of applications, including the early detection of forest fires [25], monitoring the occupation of boat spots in a Danish harbor [26], e-health [27], and agriculture [28,29]. Other IoT technologies can also be used to operate low-power wide area networks (LPWAN). For instance, SigFox technology (Sigfox, Labège, France) has recently been chosen to operate a large and low-cost radon monitoring system in Spain [30]. Besides traditional LPWAN technologies like LoRa and SigFox, the mobile industry is developing IoT solutions based on existing mobile networks (e.g., via NB-IoT standards) that are characterized by increased bandwidths [31]. Even if the coverage of public networks provided by paid services like SigFox or mobile companies solutions is constantly growing, it is mainly focused on urban, densely inhabited areas and remains inconsistent or even absent from remote and especially isolated areas. Therefore, installing a private network based on open-source LoRa technology is a reliable option for volcano observatories located in such areas. 

The deployment of IoT wireless sensor networks on active volcanoes is facing a number of challenges, some of them being already well-identified [32]: -The signal quality between gateways and nodes to collect sensor data.-The access to the internet required for data transfer from the gateway to the end point where it will be analyzed.-The energy supply needed for several months of autonomy.-The extreme conditions related to meteorology (very strong winds, heavy snowfalls, and extremely low temperatures) and, eventually, to volcanic activity (acidic plumes, damage due to ash, and bombs).

All these challenges have in practice considerably limited the adoption of IoT technologies to monitor active volcanoes. 

For all these reasons, most of the cases documented in the literature are exciting proofs of concept that were dismantled after a couple weeks or months (see, for instance, [21,23]). In the next section, we will see how we successfully circumvent the different challenges on Mt. Etna. 

### 1.4. Data Management 

Once the data are collected on the volcano and sent to the internet, the next step is to collect, store, and process them in order to make them available to the volcano surveillance experts. Some volcanoes located close to urban areas and having an historical record of deadly eruptions have dedicated observatories and already dense networks of sensors to closely monitor their activity. However, other dangerous volcanoes, such as the majority of those situated in developing countries, do not benefit from such surveillance infrastructures. As a consequence, the data cannot be stored and analyzed locally, but they can be collected remotely and shared through the internet with experts in charge as soon as they have been processed.

The chain from the sensors to the expert can be described as follows:-The sensor data are transmitted locally to a gateway using an ultra-low bandwidth local network.-The gateway transmits the data to a server through the internet using a M2M SIM card.-The data are collected, stored, and processed in a data center.-They are exported back from the data center to local or international experts for interpretation and decision-making.

The data center should be equipped with the relevant tools to automatically collect, process, visualize, and share the data (see, for instance, [33]). Due to the multiplicity of sensors of interest in volcano monitoring, the data center must also provide flexibility on data types and formats. The specific features of the environmental information have been identified for many years, as well as the inadequacy of traditional database management systems to deal with this kind of information and, particularly, to handle sensor networks data and to process data of multiple (and sometimes, unknown or fuzzy) types [34]. In these respects, many challenges for environmental data science have been recently identified, which include the problematics of uncertainty and concealing different spatial and temporal scales [35].

A data lake structure is very well-fitted to respond to these needs [36]. Data lakes are designed to store important amounts of data whatever their structure and format, from structured databases (e.g., relational databases) to semi-structured data (e.g., data in JSON format produced by sensor networks) or unstructured data (e.g., individual images or documents). Data lakes are also characterized by tools designed to allow efficient data indexing and searching. The data lake concept has emerged very recently and is just starting to be comprehensively analyzed and conceptualized [37]. We will detail in the next chapter the architecture of the data lake that was created to collect the data produced by the sensors deployed on Mt. Etna.

## 2. Materials and Methods

This section is dedicated to a description of the different elements that were designed and deployed to enable the continuous monitoring of airborne radon on Mt. Etna. This section will highlight the innovative aspects of our work, especially:-The tropicalization of the radon probes to protect them from high-altitude mountain conditions and volcanic plumes.-The use of versatile LoRaWAN nodes dedicated to environmental instrumentation.-The sensor network infrastructure on-site for data collection and transfer.-The data lake architecture in Clermont-Ferrand University Data Center.

### 2.1. Tropicalization of the Radon Probes

We chose the ÆR+ radon probe from Algade (Bassines-sur-Gartempe, France; [17]) for its very good compromise between sensitivity (about 20 Bq/m^3^ per hit per hour) and autonomy. Moreover, the radon measurement performed by ÆR+ is in compliance with the norm ISO 11665-5 (measurement of radioactivity in the environment—Air: radon-222—Part 5: Continuous measurement methods of the activity concentration) [38]. The probe was originally designed for continuous indoor airborne radon measurements. To adapt it to outdoor activity measurements, a list of requirements was produced that resulted in a number of improvements illustrated in Figure 1: -As radon measurements are affected by the temperature and humidity in the probe diffusion chamber, a 1-W heating resistance was added to maintain the temperature above 15 °C and humidity below saturation. As ÆR+ also measures continuously the temperature and humidity, the operation of this heating system is internally controlled by the ÆR+ itself.-The ÆR+ probe was installed in a polystyrene box (see Figure 1) to increase thermic insulation. The polystyrene box itself was installed in a larger metal box containing also a 3.2 V/180 Ah LiFePO_4_ battery to restore the probe autonomy despite the addition of the heating system. An M12 connector was installed to transfer ÆR+ data to a LoRa module (described hereafter).-Two air entries were opened into the polystyrene box where the ÆR+ is placed (one on the side and the other one on the bottom of the main box). This configuration allows fast, passive circulation of air around the detector. Moreover, it is also meant to preserve the connection of the detector with outdoor air if one of the entries is obstructed, for instance, because of heavy snowfalls (Figure 2f).-For the station to be deployed on the active crater rim of Mt. Etna (see next paragraph), the ÆR+ sensor was wrapped into a thin film of parafilm, allowing the diffusion of radon but protecting the sensor from acidic gases.-A Universal Asynchronous Receiver Transmitter (UART) interface is used for the data readout.

Two radon probes were permanently installed on the South side of Mt. Etna Volcano in September 2019. One station was installed on top of the Mt. Frumento Supino cone (see Figure 2d and Figure 3), and a second one was installed on the oriental flank of the Mt. Barbagallo cone (also called the 2002–2003 crater; see Figure 2e,f and Figure 3). The first site is located under the wind and is likely to be representative of the local atmospheric background. The second site is located on the flank of a cone presenting a peculiar hydrothermal system where very high soil emissions (including radon) have been noticed in the past [39,40]; therefore, it was chosen to study the potential influence of soil radon emissions on air activity. Another station was also deployed on the Western part of the Bocca Nuova crater rim on top of the volcano from June to October 2019 in order to evaluate the quality of the tropicalization in an acidic environment. This station was removed in October 2019 due to threatening strombolian explosive activity in the nearby Voragine Crater.

### 2.2. LoRaWAN Network, Communicating Nodes, and Gateways 

LoRa technology was chosen to connect the sensors. The choice of LoRaWAN (LoRa wide-area network) was guided by the advantages of private networks compared to public networks in the context of volcano monitoring:-Private network infrastructures can be installed by volcano observatories to fit their specific needs and constraints, while public networks rely on infrastructures that are managed independently by private companies and cannot be easily modified.-Financial investments to operate a private network (hardware and maintenance) is more favorable than a subscription to paid public networks when a long-term operation is planned and when the number of connected sensors is high, which is the case of volcano observatories.

Finally, since the monitoring chain we deployed is aimed at being transferred to other volcanoes, especially in developing countries, it was also preferred to adopt solutions with minimal dependence on external factors like private companies.

The LoRaWAN network was installed in September 2019 in order to transfer radon measurements from the sensor to a server located in France. The network topology deployed on Mt. Etna is displayed in Figure 4. The two radon stations are connected to end nodes that transfer the data to a gateway. The gateway collects all data received from the two nodes and sends them to the server. The communication between the nodes and the gateway is performed using the LoRaWAN Protocol [41] and the LoRa physical layer. This technology allows communication over long distance (several kilometers) but with limited message sizes (230 bytes at most). LoRaWAN communication uses license-free sub-gigahertz radio frequencies available in Europe (868 MHz with a bandwidth of 125 kHz). The gateway (Wirnet IoT Station developed by Kerlink, Thorigné-Fouillard, France; see Figure 2c) acts as a bridge between the nodes and the LoRaWAN server located hundreds of kilometers away in Clermont-Ferrand, France using the Packet Forwarder software from Semtech company (Semtech, Camarillo, CA, USA). This gateway is connected to the server through the internet network using 3G mobile telecommunication thanks to an international multioperator SIM card purchased to the Things Mobile IoT company (Things Mobile, Milano, Italy). The server is based on the components provided by ChirpStack, an open-source LoRaWAN Network Server stack.

A third node was also installed in a stand-alone configuration on Montagnola Peak at a short distance from the gateway but behind an important rock obstacle in order to assess the quality of transmission in this particular configuration. Communication tests were also performed from Mt. Etna Summit craters (Bocca Nuova; see Figure 3). 

The communicating LoRa end nodes (see Figure 2b) were developed within the context of a regional project in Auvergne, France. They were designed to allow data reading from a large variety of associated sensors. Each node has an internal memory to store data read from the sensor output and is able to send data to the gateway at a programmable time frequency. Messages sent by the node are called frames and contain a few measurement results. Due to the potential loss of communication between the node and the gateway or between the gateway and the server, frames sent by the node can be lost before reaching the server. In order to minimize the power consumption, the communicating node only makes one transmission attempt to the server per awakening period. If the node does not receive an acknowledgement signal from the server, it considers the transmission has failed and goes back to sleep mode. At the next scheduled time of transmission, the node sends in priority the newly acquired data frames to the server. Within the limit of the payload, it also sends the frames it failed to transmit previously. If several frames are to be retransmitted due to a long interruption of communication with the server, the node is programmed to resend in priority the most recent one in order to fit the needs of real-time monitoring and not to resend frames older than 7 days in order to spare the memory space. 

Different modulations of the LoRa signal can be selected in order to improve the signal-to-noise ratio and, therefore, the range of transmission, but this implies limiting the data rate (the size of the data frame). In the present deployment, the data rate was fixed at 5 (the spreading factor being set to 7) in order to operate with the maximum allowed payload.

Regarding autonomy, the power supply of the node is separated from the power supply of the radon sensor and is made of a single 3.6 V/9 Ah Li-Ion battery. All communication nodes are also equipped with small sensors recording temperature, atmospheric pressure, relative humidity, and node battery voltage. Data from these internal sensors can also be sent to the server in order to monitor the autonomy of the node and the meteorological conditions around it. 

The performances of this node coupled to the capabilities offered by the LoRaWAN technology offer a large flexibility on the type and number of sensors that can be integrated in the network. Moreover, the sensors can be deployed up to several kilometers away from the gateway and for several months thanks to the node low consumption.

The choice of the gateway location is the result of a compromise between the quality of the 3G network, the LoRa signal range from the stations, and the necessity to protect the gateway from winter extreme conditions. Due to the volcano topology and continuous activity of its summit craters, no single location allows a complete coverage of the areas of interest. After extensive tests conducted in July 2019, the gateway location was chosen in the INGV (Istituto Nazionale di Geofisica e Vulcanologia) instrumental shelter near Montagnola Peak (see Figure 3), about three kilometers south of the Etna Summit craters. The building did provide protection against winter storms and an energy supply but reduced the LoRa transmission range because of the RF signal screen induced by the building’s metallic walls. 

### 2.3. Data Lake

Figure 5 displays the structure of the data lake designed to respond to the needs of the academic research community to collect, store, and display environmental data coming from connected objects. The data packets are sent by the different LoraWAN networks through the internet and ingested as Json files into the data lake using the Elastic Stack Suite (Beats, Logstash and Elasticsearch products from Elastic, Mountain View, CA, USA; [42]) used here in the context of volcanic surveillance for the first time. As shown in Figure 5, Beats is a platform comprising the data shippers that transfer the data from collection points (the LoRa server, in our case) into the data lake. Logstash performs data ingestion into the data lake (logging of data received from Beats, labeling and sending to storage). Elasticsearch is a search engine able to work in the data lake. It can be coupled to the graphical interface Grafana (Grafana Labs, New York, NY, USA; [43]) for real-time data visualization on dashboards, automatic monitoring, or alarm triggering. Data can be also exported outside of the data lake using a Message Queuing Telemetry Transport protocol (MQTT) to share them with data users [44]. 

The data packets are stored for later usage either as flat files or in a relational PostgreSQL [45] database, where their consistency is checked. A website allows data lake users to enrich their datasets with metadata, while publication tools are made available, including a GeoNetwork data catalog [46].

## 3. Results

During six months from October 2019 to March 2020, the performance of the sensors network installed on Mount Etna was evaluated. This network operated in complete autonomy with autumnal and winter weather. 

### 3.1. Autonomy and Mechanical Integrity of the Stations

#### 3.1.1. Autonomy

Except the gateway, all the equipment deployed on Etna is power-supplied thanks to batteries (see Materials and Methods). Expected autonomy for both radon sensors and communicating nodes was estimated to be of several months (at least six months) but can eventually be reduced due to low temperatures decreasing the battery capacities. Data collected during the first six months of deployment of these devices give us an estimation of the lifetime of the batteries in field conditions.

First of all, the battery voltage of the radon stations has been monitored. At the time of writing, no significant deviation of battery capacity has been observed due to winter conditions, and batteries are progressively discharging.

Regarding communicating nodes, it was possible to observe very different discharge curves between the nodes. Indeed, the node configuration is expected to have a direct impact on its autonomy. Sensor reading frequency, data transmission periodicity, and LoRa data rate (modifying the spreading factor and the bandwidth) are likely to influence the activity of the device and, therefore, the final autonomy of the node. Battery voltages of LoRa nodes have been continuously monitored and transmitted to the LoRa server. Battery voltage evolution through time for nodes located at Mt. Barbagallo and Montagnola is reported in Figure 6 over a three-month period (Figure 6a), with a specific focus on the first 35 days (Figure 6b) where linear regression models were applied to fit the data. First of all, it should be pointed out that this monitoring has allowed the detection of an early battery discharge (Figure 6a), and an anticipated maintenance operation has been carried out on-site in order to change the faulty component. It can be observed that the battery discharge is 1.6 times faster for the node located on Mt. Barbagallo compared to the node located in Montagnola (2.22 mV per day against 1.36 mV per day) over the first 35 days of operation. The two nodes operate at different frequencies: the node located on Mt. Barbagallo is awake every 15 min to acquire measurements from sensors and send data to the LoRa server, whereas the node at Montagnola is awake once every hour. However, the difference in battery discharge could also be due to different discharge curve shapes between the batteries, related, for instance, to different histories of charge/discharge or temperature effects. To further investigate this issue, current absorption measurements are planned. 

Finally, it should be pointed out that the flexibility of the communication node in terms of both software and hardware allows replacing batteries with higher capacity models or adjusting the set-up (reading, transmission, and data rates) in order to reach a given autonomy. 

#### 3.1.2. Mechanical Integrity

A visit to the stations on 31st of December 2019 allowed checking their mechanical integrity during winter. Temperatures inside the stations were recorded to vary between −14 °C and 20 °C since their deployment. Despite these large temperature variations, no damage to the sensors and the nodes was noticed. Only the presence of a thick layer of ice around the Mt. Barbagallo Station was observed, as shown in Figure 2f. However, it did not alter the LoRa communication (see after).

The radon probe that was installed on the Bocca Nuova Crater rim under the influence of volcanic acidic plume did work properly during 2.5 months from July to mid-September 2019, and when it was dismantled, we did not notice significant damage of the ÆR+ sensor that was protected within a plastic film (see Methods).

### 3.2. Performances of the Wireless Network

The performances of the wireless data transmission have been evaluated over the period October to December 2019. Table 1 displays the data loss rate (percentage of data never received by the server) and the first-try transmission failure rate (percentage of frame transmissions that failed at the first try) for the three communicating nodes. The data loss rate is about 2.7% at Mt. Barbagallo Station and about 1% at Monte Frumento. The degraded performance at the Montagnola site (37% loss) was expected, as the communication node was deliberately located behind a topographic obstacle for testing purposes. It is interesting to notice that the first-try transmission failure rate is significantly higher than the data loss rate (about 10% for Mt. Barbagallo and Mt. Frumento and about 50% for Montagnola). This means that the system in charge of detecting transmission failures has been properly working, allowing some data to be resent later and finally registered by the server.

The overall transmission rate varies between 97% and 99% for obstacle-free configurations. However, this figure does not inform about the continuity of transmissions, since failed transmissions are reprogrammed later and finally received by the server. This transmission continuity is extremely important for near real-time monitoring of the volcano activity. In order to assess it, the cumulative number of frames received by the server over a period of 3.5 days is represented in Figure 7. The main trend observed at the beginning and at the end of the period is a linear increase corresponding to the normal rate of frame transmission (one every 15 min). Starting from 38.3 days until 39 days, however, the server does not receive any more data, indicating a temporary loss of transmission. When the transmission is back, the cumulative number of frames increases again but at a higher rate, until it reaches the main trend curve 1.3 days after. This feature corresponds to a phase of more intensive transmissions, because frames that could not be sent during the interruption were progressively resent along with new measurements. This illustrates the efficiency of the failure detection system that allows data to be recovered later after an interruption. 

The actual delay between the time a measurement is made, and the time when it is registered in the Clermont-Ferrand data lake is shown in Figure 8. It can be seen that 94% of all measurements are transmitted within one hour to the server, while some measurements can be received as late as 100 h after they were made. The approximately flat distribution between 1 and 100 h is due to the fact that most of the transmission interruptions were not isolated but rather grouped in periods extending from a few hours up to a few days (possibly due to persisting regimes of unfavorable meteorological conditions). The high percentage of measurements transmitted within one hour confirms that the LoRa network is very adapted to near real-time data transfer in the context of volcano monitoring.

### 3.3. Preliminary Radon Measurement Results

Figure 9 displays almost one month of data recorded at Mt. Barbagallo and Mt. Frumento Stations from 25/10/19 to 21/11/19. Measurements were performed every 15 min by ÆR+. Due to low activities in the air, measurements performed every 15 min are dominated by a statistical noise due to random fluctuations of the signal. In order to remove this noise, the running 48-h average of activity is presented for both stations. Radon activity at Mt. Barbagallo varies around a mean value of 83 ± 0.4 Bq/m^3^, while at Mt. Frumento, the activity has a mean value of 63 ± 0.3 Bq/m^3^ (2σ uncertainty on the mean under the Poisson statistics assumption). These activity levels are significantly different but are in qualitative agreement with measurements previously reported [7,8]. Higher radon activity at Mt. Barbagallo can be explained by the presence of an active hydrothermal system under the cone, supported by many geological evidences ([39,40]), while such a system is clearly absent at Mt. Frumento, which is a much older and extinct edifice [47]. Moreover, some significant variations of the radon activity in the air are observed at both stations, which confirms the suitability of ÆR+ sensors to record air radon activity changes on a timescale of a few days. Even if a statistical study to fully exploit this dataset in relation with meteorological and volcanological information is out of the scope of the present study, these preliminary results demonstrate the feasibility of radon monitoring in the air on volcanoes with low-cost sensors like ÆR+.

## 4. Conclusions

In this paper, we have presented the first complete implementation of a wireless network of outdoor air radon probes on the Mount Etna Volcano in Sicily. This was only made possible thanks to a number of specific developments: the tropicalization of a radon probe; the installation of a LoRa network with flexible and robust end nodes; and the construction of a data lake for sensor data collection, storage, and analysis. 

Sensor tropicalization has proven very efficient, as airborne radon probes remained operational during winter and in the presence of volcanic plumes. The installed LoRa network allows several months of autonomy with a global transmission rate close to 100% and a very short delay of transmission (90% of the data are transmitted within 15 min). Further tests are needed to evaluate how the node autonomy is impacted by its configuration.

The network is currently operational on Mt. Etna and continues to transmit data. The gateway is protected from meteorological conditions, as it is installed in the INGV building at Montagnola, but this reduces its range. We are exploring different options to increase this range by implementing a robust outdoor LoRa antenna. The other foreseen improvement will be next year to install additional airborne radon probes, especially on the summit crater rim, but also to couple radon activity measurements in the air to measurements in the soil. 

At the other end of the pipeline, the data lake has been collecting data in a very steady way since the day the gateways were turned on. A number of enhancements are foreseen to improve user friendliness and data security.

Our results overall demonstrate that wireless sensor networks based on LoRaWAN technology are relevant for near real-time surveillance of active volcanoes. All components of the architecture we have deployed, from the communicating nodes to the data lake software stacks, use open-source technologies. The communicating nodes are flexible and can be interfaced to other sensors to measure temperature, position, humidity, deformation, tilt, etc. The only limitation is the ultra-low bandwidth that makes this architecture unsuitable for large data transfers, such as those required by imaging or video monitoring. 

Our solution is easy to deploy and cheap. The gateway connection to the internet just requires purchasing a SIM card. Developing countries have wide mobile coverage—in particular, in the populated zones close to the dangerous volcanoes that require constant monitoring. We foresee extending our tests to such volcanoes in the coming months. 

## Figures and Tables

**Figure 1 sensors-20-02755-f001:**
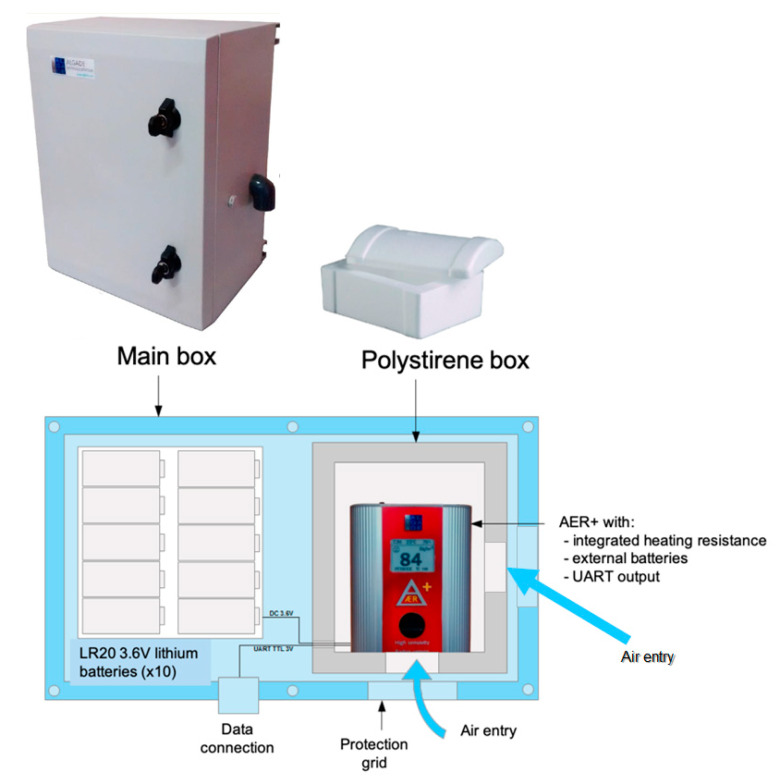
Tropicalization of ÆR+ radon probe by Algade. The dimensions of the main box are 400 × 300 × 210 mm.

**Figure 2 sensors-20-02755-f002:**
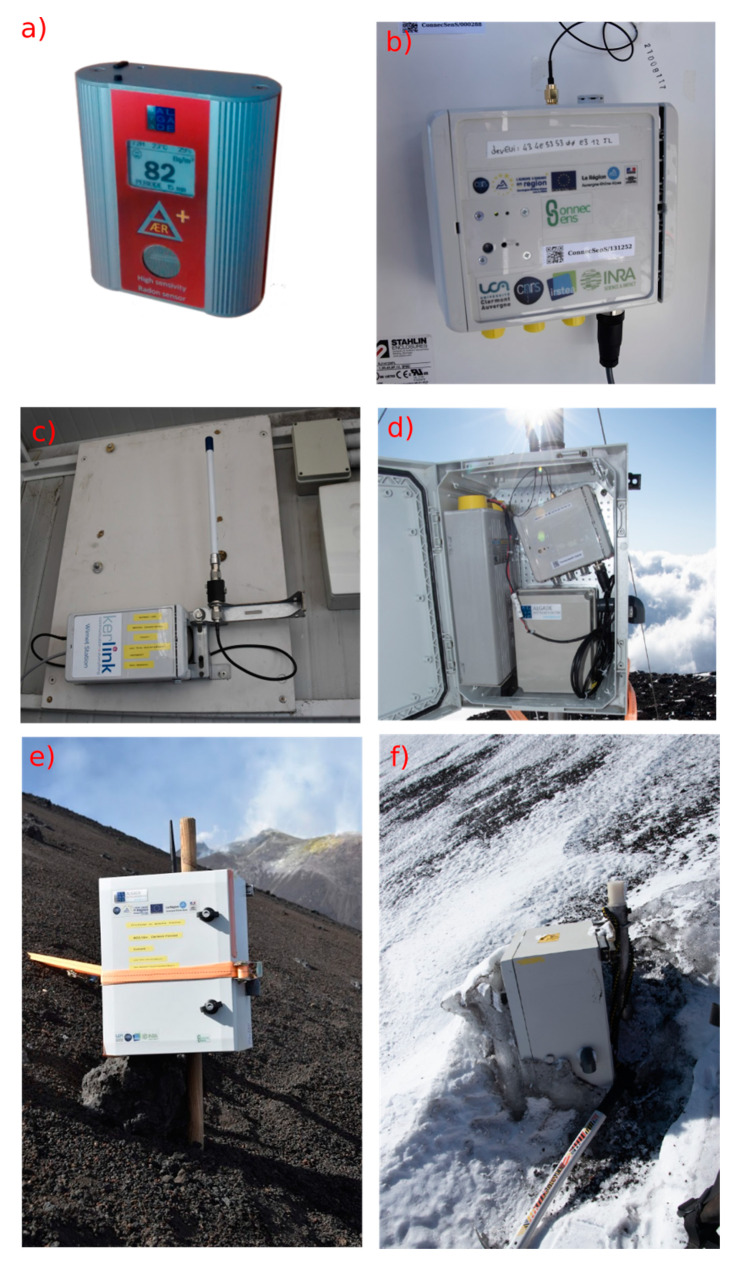
Different views of the equipment installed on Mt. Etna: (**a**) the ÆR+ sensor; (**b**) the communicating node; (**c**) the gateway in Montagnola Shelter; (**d**) the radon station on Mt. Frumento Supino comprising the ÆR+ box, the battery, and the communicating node; (**e**) the radon station on Mt. Barbagallo in September 2019; and (**f**) the same station in December 2019.

**Figure 3 sensors-20-02755-f003:**
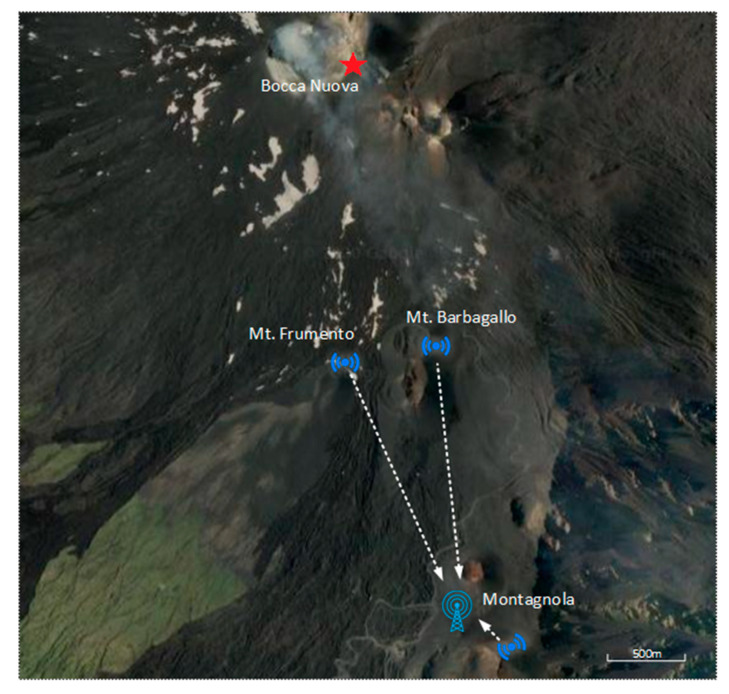
Map of the network with positions of sensor-attached nodes (Mt. Frumento and Mt. Barbagallo), gateway (Montagnola Shelter), and stand-alone node (Montagnola Peak). The distance between the gateway and the radon stations (Mt. Frumento and Mt. Barbagallo) is about 1.7 km. Communication with the gateway has also been tested from the Bocca Nuova Crater rim (red star).

**Figure 4 sensors-20-02755-f004:**
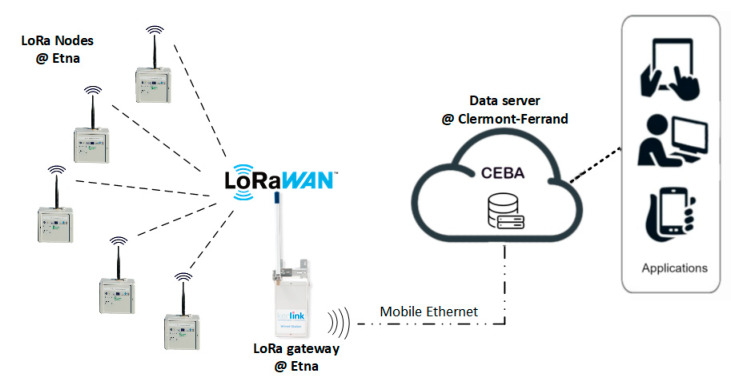
LoRaWAN (wide-area network) topology deployed from the LoRa nodes to the server (called CEBA).

**Figure 5 sensors-20-02755-f005:**
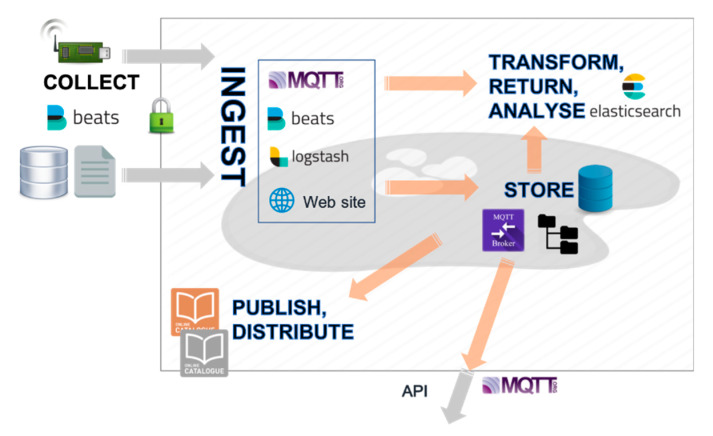
Structure of the data lake implemented in the Université Clermont Auvergne Mésocentre. Beats sends data to Logstash, an open-source data ingestion tool. Data can be retrieved and analyzed using the Elasticsearch engine and visualized on real-time dashboards or exported outside of the data lake using MQTT.

**Figure 6 sensors-20-02755-f006:**
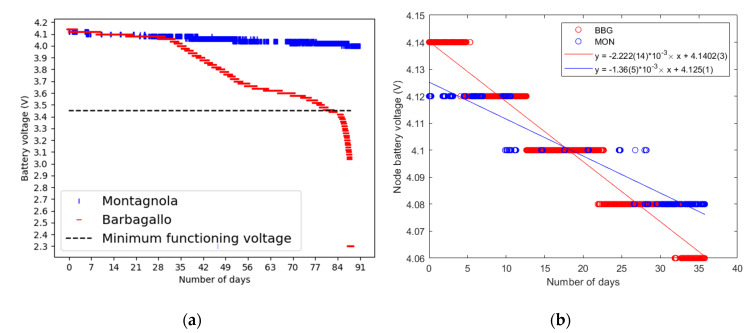
Variation of the battery voltages with time for nodes at Montagnola (MON) and Mt. Barbagallo (BBG) during a 3-month period (**a**) with a focus over the first 35 days at the onset of the deployment (**b**). Figure 6b also displays linear regression models together with the corresponding coefficients and their standard errors for a 95% confidence interval.

**Figure 7 sensors-20-02755-f007:**
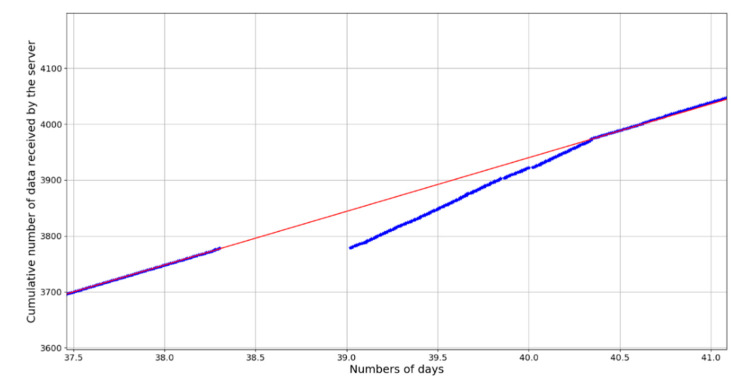
Cumulative number of data frames received by the LoRa server (from a Mt. Barbagallo node) before, during, and after an interruption of communication of about a half-day. The red linear trend represents the data production rate computed from the number of active sensors associated to the node and their frequency of acquisition. After the interruption, the rate at which data are transmitted to the server increases in order to transfer the data acquired during the interruption until it comes back to the normal data production rate curve after ~2.5 days.

**Figure 8 sensors-20-02755-f008:**
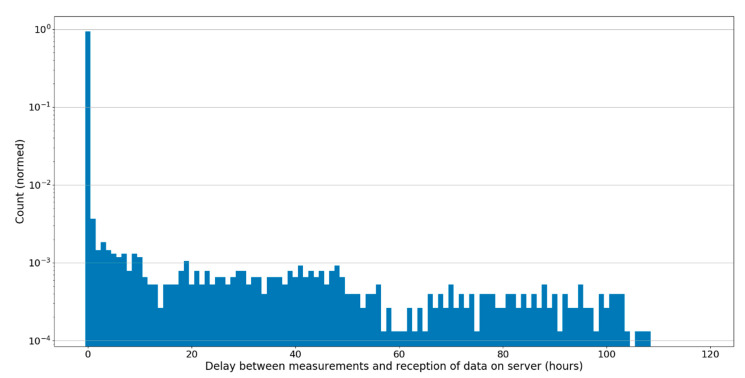
Logarithmic distribution of the delay between measurement and registration in the server (hours) for the Mt. Barbagallo Station. Most of the measurement results (94%) are transferred to the server within one hour.

**Figure 9 sensors-20-02755-f009:**
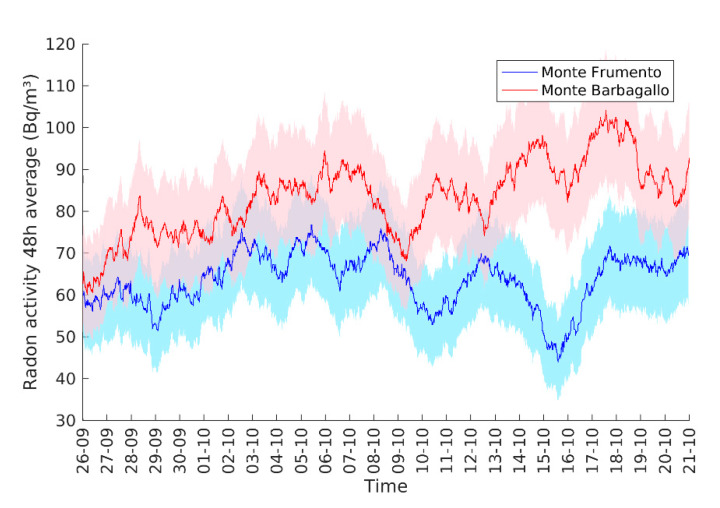
Running a 48-h average of radon activity measured at the Mt. Barbagallo and Mt. Frumento Supino Stations on the Mt. Etna South flank. Colored zones corresponds to a 2-sigma uncertainty in the 48-h mean (computed under the Poisson statistics hypothesis).

**Table 1 sensors-20-02755-t001:** Transmission statistics table. The data loss rate corresponds to the global amount of data not transferred to the server, while the frame-first transmission failure rate is the observed probability of the system to fail the first transmission of a frame. The significance of the loss rate is computed from the number of missing measurements under a Poisson statistics assumption (relative standard deviation at 2σ is $2/\sqrt{N}$).

Node Location	Data Loss Rate	Statistical Significance (2σ)	Frame-First Transmission Failure Rate
Mt. Barbagallo	2.7%	±0.4%	9.2%
Mt. Frumento	1%	±0.4%	13.1%
Montagnola Peak	37%	±2.5%	54%

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
