# Peer review of "From Sensor to Cloud: An IoT Network of Radon Outdoor Probes to Monitor Active Volcanoes"

_sensors, 2020, doi:10.3390/s20102755_

Round 1

Reviewer 1 Report

The paper is well written and describes an interesting experiment.

Some issue

1) The retransmission mechanism implemented on top of LoraWan is not adequately described.

2) Lorawan configuration parameters should be added (e.g. frequency, sf, ack slots, etc..). In particular, considering the parameters involved in the retransmission protocol and in the battery consumption. 

3) In the result section, the authors justify the different discharge curves "by the difference of their respective frequency of activity". I have some doubts about their conclusion and would ask them for further investigations on the point. would it be possible to measure the current absorbed in the two configurations? when does the knee of the second configuration take place?

Reviewer 2 Report

Author present in a clear way their work. Graphics are self explicative too, altought the size of some of them (e.g. figure 1 or figure 5) should be reduced.

Regarding LoRaWAN network, there is no info related with other techonologies either tested or studied for authors, like Sigfox, NBIoT or others (i.e. Alvarellos, A.; Gestal, M.; Dorado, J.; Rabuñal, J.R. Developing a Secure Low-Cost Radon Monitoring System. Sensors 202020, 752.)

Reviewer 3 Report

This paper thoroughly describes the design and implementation of a radon radon sensor network for volcano monitoring using a LPWAN network, as well as a data management system for storage and analysis. 

The topic is novel, and the technology selection and implementation are really interesting. Moreover, the paper is well structured and very descriptive. 

would like to make only a few questions or comments to help further improve the quality of the work. 

First of all, I miss some more detailed information about the deployed LoRaWAN network, the gateway selection, characteristics, which type of Lora network you used (private, public) and the specific server solution used. 

Also, about the gateway, you mention that the range is limited by the metallic walls of the building. Did you considered using an external antenna outside of the building while keeping the electronics inside? 

About the analysis of lost packets, you mention the percentage of packets lost after retransmission. However, in Figure 7 looks like the system is trying to retransmit lost packets during a long period of time. When is a packet considered lost? how many times does it try to retransmit before it's considered a failure? Also are the lost packets randomly distributed over time or are they grouped due to long interruptions like in figure 7? 

In figure 2f it is shown one of the sensors covered in snow specially on the bottom part where is the opening for air entry. You mention that the Lora communication was still working properly, but, are the radon measurements accurate even in those conditions? 

In figure 6 the battery consumption of both nodes at the beginning seems to be almost identical, however after day 35 Barbagallo node starts to decrease voltage much faster than Montagnola. Is there a reason for this abrupt change on day 35? Also does the node still work below the functioning voltage? what is the reason of the last erratic measurements? 
